

# Devil's staircase of topological Peierls insulators and Peierls supersolids

Titas Chanda[1,2⋆], Daniel González-Cuadra[3,4,5†], Maciej Lewenstein[5,6], Luca Tagliacozzo[7,8] and Jakub Zakrzewski[2,9]

**1** The Abdus Salam International Center for Theoretical Physics, Strada Costiera 11, 34151 Trieste, Italy
**2** Instytut Fizyki Teoretycznej, Uniwersytet Jagielloński, Łojasiewicza 11, 30-348 Kraków, Poland
**3** Center for Quantum Physics, University of Innsbruck, 6020 Innsbruck, Austria
**4** Institute for Quantum Optics and Quantum Information of the Austrian Academy of Sciences, 6020 Innsbruck, Austria
**5** ICFO-Institut de Ciències Fotòniques, The Barcelona Institute of Science and Technology, Av. Carl Friedrich Gauss 3, 08860 Barcelona, Spain
**6** ICREA, Passeig Lluis Companys 23, 08010 Barcelona, Spain
**7** Instituto de Física Fundamental IFF-CSIC, Calle Serrano 113b, Madrid 28006, Spain
**8** Departament de Física Quàntica i Astrofísica and Institut de Ciències del Cosmos (ICCUB), Universitat de Barcelona, Martí i Franquès 1, 08028 Barcelona, Catalonia, Spain
**9** Mark Kac Complex Systems Research Center, Uniwersytet Jagielloński, Kraków, Poland

⋆ tchanda@ictp.it, † daniel.gonzalez-cuadra@uibk.ac.at

## Abstract

We consider a mixture of ultracold bosonic atoms on a one-dimensional lattice described by the XXZ-Bose-Hubbard model, where the tunneling of one species depends on the spin state of a second deeply trapped species. We show how the inclusion of antiferromagnetic interactions among the spin degrees of freedom generates a Devil's staircase of symmetry-protected topological phases for a wide parameter regime via a bosonic analog of the Peierls mechanism in electron-phonon systems. These topological Peierls insulators are examples of symmetry-breaking topological phases, where long-range order due to spontaneous symmetry breaking coexists with topological properties such as fractionalized edge states. Moreover, we identify a region of supersolid phases that do not require long-range interactions. They appear instead due to a Peierls incommensurability mechanism, where competing orders modify the underlying crystalline structure of Peierls insulators, becoming superfluid. Our work show the possibilities that ultracold atomic systems offer to investigate strongly-correlated topological phenomena beyond those found in natural materials.



# 1 Introduction

In the last decades, synthetic quantum systems such as ultracold gases, trapped ions, superconducting qubits or photonic systems have been employed as quantum simulators [1] to investigate models from condensed matter and high-energy physics [2–7], allowing to tackle open problems in these areas beyond the capabilities of classical computers [8]. Ultracold atoms in optical lattices, in particular, present a perfect playground to investigate quantum many-body phenomena under controllable experimental conditions [9–11]. They offer the possibility to engineer a broad range of interactions [12–16], leading to the preparation of many different synthetic phases of matter, from supersolids [17–20] to topological quantum matter [21–25], in many cases without known counterparts in natural materials [26].

The degree of control achieved, in particular, with mixtures of bosonic atoms has enabled implementations of various types of correlated tunneling terms, leading to the quantum simulation of lattice gauge theories [27–30], as well as spin-exchange interactions, giving rise to antiferromagnetic states [31, 32]. Motivated by these recent breakthroughs, we consider a bosonic mixture in a one-dimensional lattice which, in the appropriate limit, is described by a non-standard Bose-Hubbard (BH) model that combines these two elements. More specifically, we first consider a bosonic species described by the standard BH model. A deeply-trapped second species is then introduced, possessing two internal states described in terms of spins. By tuning appropriately the experimental parameters, an effective model with correlated tunneling emerges. The resulting Hamiltonian, known as the $\mathbb{Z}_2$ Bose-Hubbard ($\mathbb{Z}_2$BH) model, was first introduced to simulate dynamical lattices in ultracold atomic systems [33], where spin degrees of freedom play a similar role to phonons in solid-state systems. In this case, the dynamical lattice interact with bosonic matter, giving rise to bosonic versions of the Peierls mechanism [34] and Peierls insulators with bond order wave (BOW) order at certain fillings, akin to the fermionic Su-Schrieffer-Heeger (SSH) model [35]. Similarly to the latter, the Peierls insulators found in the $\mathbb{Z}_2$BH model present non-trivial topological characteristics induced by interactions [36, 37]. They are, in particular, symmetry-breaking topological insu-

lators, where BOW long-range order coexists with symmetry-protected topological properties. Among other features, the interplay between symmetry breaking and symmetry protection gives rise to dynamical symmetry-protected topological defects and to the fractionalization of bosonic matter [38, 39].

In this work, we take advantage of the flexibility that cold-atomic systems offer and show that, apart from their role as quantum simulators of solid-state systems, they can be used to explore novel strongly-correlated phenomena beyond those found in natural materials. In particular, here we introduce antiferromagnetic interactions between spins in the $\mathbb{Z}_2$BH model and analyze the resulting XXZ-Bose-Hubbard (XXZ-BH) model. First, we show how a Devil's staircase structure [40–42] of Peierls insulators appears in the parameter space. Every step presents a different type of BOW order and, moreover, every one of them is a symmetry-protected topological phase. These results extend, therefore, the previously studied topological Peierls insulators [38, 39] to every possible rational value of the bosonic density. Devil's staircase structures have been studied in long-range interacting models [43–47], e.g., via long-range dipolar interactions among particles. Here, we report such structures in a nearest-neighbor interacting model.

Furthermore, the competition between the correlated tunneling and the spin interactions gives rise to supersolid (SS) phases, where BOW long-range order coexists with superfluidity. SS was first considered as a quantum phase for Helium-4 [48–50]. However, despite some experimental evidences [51], there is still an ongoing debate about the SS character of its ground state [52]. More recently, it was proposed that SS may be realized with trapped ultracold atoms or molecules with dipolar interactions [17, 53–55] or Rydberg-dressed atoms [56]. SS may be also induced dynamically in systems with short range interactions as an out-of-equilibrium state [57] or in Bose-Fermi mixtures [58–60]. Recently, supersolidity was observed in different atomic experiments: with cold atoms coupled to a cavity mode [19], in the presence of spin-orbit coupling [20], and in a dipolar Bose-Einstein condensate [61]. Here, we uncover a different mechanism to obtain SS phases, requiring only short-range interactions and bosonic atoms. We recognize how a *Peierls incommensurability* takes place due to the competition of various types of long-range order patterns, generating a region of Peierls SS phases. This mechanism is different from the more standard crystal melting that creates SS phases in long-range interacting bosonic [43] and spin systems [44, 62, 63].

The rest of the paper is organized as follows. In Sec. 2, we introduce the XXZ-BH model and summarize our main results. These are then discussed in detail in Sec. 3. Specifically, in Sec. 3.1 we introduce the bosonic Peierls mechanism, central to describe the physics of the model. In Secs. 3.2-3.4, we analyze the staircase structure of the commensurate order and the region with incommensurate order that appear in the system, including topological Peierls insulators and supersolids. In Sec. 4, we describe a quantum-simulation scheme for the model, showing how it emerges as the effective description for a mixture of ultracold bosons in optical lattices. Finally, we summarize our conclusions in Sec. 5.

# 2 XXZ-Bose-Hubbard chain

## 2.1 The model

We consider a bosonic system on a one-dimensional lattice described by the following Hamiltonian,

$$\hat{H}_{\mathbb{Z}_2\text{BH}} = -\sum_{i=1}^{L-1} \left[ \hat{b}_i^\dagger \left( t + \alpha \hat{\sigma}_i^z \right) \hat{b}_{i+1} + \text{H.c.} \right] + \frac{U}{2} \sum_{i=1}^{L} \hat{n}_i(\hat{n}_i - 1) - \mu \sum_{i=1}^{L} \hat{n}_i, \tag{1}$$

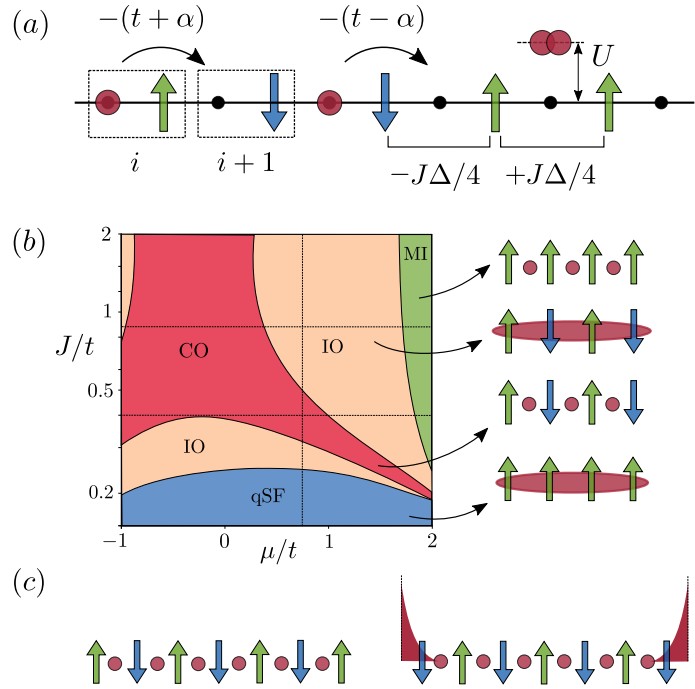

Figure 1: **The XXZ-Bose-Hubbard model: (a)** sketch of the system described by Eqs. (1)–(3) in the limit $\Delta \gg 1$. Bosonic particles (red spheres) can tunnel between the nearest-neighbor sites of a one-dimensional chain, with tunneling elements that depend on the state of spin-1/2 systems located at the bonds (arrows). The bosons interact on-site according to the Bose-Hubbard model, and the spins do so via a XXZ interaction. **(b)** Qualitative phase diagram of the model in terms of the chemical potential $\mu$ and the spin interaction $J$. Apart from the standard MI and qSF phases, there are two regions where both the spins and the bosons develop long-range order via a bosonic Peierls transition. Each of them contains a staircase of distinct phases characterized by pairs of values of $N$ and $M$. In the commensurate case (CO), these correspond to insulators, while in the incommensurate one (IO) long-range order coexists with superfluidity, giving rise to supersolids. The dotted straight lines indicate the cuts in the phase diagram along which data in Fig. 4 are presented. **(c)** Each of these phases exist for the two signs of $M$. In the CO region, positive values of magnetization $M$ correspond to trivial phases, while negative values give rise to non-trivial topological properties in the Peierls insulators as witnessed by the presence of localized edge states.

where $\hat{b}_i$ ($\hat{b}_i^\dagger$) is the annihilation (creation) operator of a bosonic particle on site $i$ and $\hat{n}_i$ is the corresponding number operator. The operators $\hat{\sigma}_i^\beta$, with $\beta = x, y, z$, are the Pauli operators for a spin-1/2 systems living on the bond between sites $i$ and $i+1$ (Fig. 1**(a)**). This Hamiltonian, known as the $\mathbb{Z}_2$ Bose-Hubbard model ($\mathbb{Z}_2$BH), was introduced to model lattice degrees of freedom on a system of ultracold atoms in an optical lattice [33]. The spins in the $\mathbb{Z}_2$BH model (1) can be interpreted as truncated phonons, interacting with (bosonic) matter particles by mediating their tunneling. This model was studied in the presence of external fields, which introduce quantum fluctuations on the spins, uncovering various solid-state like phenomena such as bosonic Peierls insulators [36, 37] and boson fractionalization induced by topological

defects [38, 39]. Here, we consider spin dynamics given by the XXZ Hamiltonian,

$$\hat{H}_{\text{XXZ}} = \frac{J}{4} \sum_{i=1}^{L-2} \left( \Delta \hat{\sigma}_i^z \hat{\sigma}_{i+1}^z + \hat{\sigma}_i^x \hat{\sigma}_{i+1}^x + \hat{\sigma}_i^y \hat{\sigma}_{i+1}^y \right), \tag{2}$$

so that the total Hamiltonian of the system is given by a XXZ-Bose-Hubbard model (XXZ-BH),

$$\hat{H}_{\text{XXZ-BH}} = \hat{H}_{\mathbb{Z}_2\text{BH}} + \hat{H}_{\text{XXZ}}. \tag{3}$$

This model possesses a $U(1) \times U(1)$ symmetry corresponding to the conservation of total particle number $N = \sum_i \langle \hat{n}_i \rangle$ and magnetization $M = \sum_i \langle \hat{\sigma}_i^z \rangle$. The interplay between these two conserved quantities will be crucial, as we shall see, to describe the different phases of the model. In the hardcore limit, with $U/t \to \infty$, the bosons can be mapped to spinless fermions via a Jordan-Wigner transformation, and the system presents an extra chiral symmetry at half filling. In Sec. 4, we present a quantum simulation scheme for the XXZ-BH model using a bosonic mixture of ultracolds in an optical lattice.

In this paper, we set $U = 20t$, $\Delta = 2$ and $J > 0$. In such a setting, the bosons are strongly repulsive and the XXZ Hamiltonian favors Néel order in the spins. We also fix $\alpha = 0.5t$ throughout the paper and consider bosonic densities $\rho = N/L \leq 1$. The ground state of the system is obtained via matrix product state (MPS) [64, 65] based density matrix renormalization group (DMRG) [66, 67] method with bond dimension $\chi = 400$ and with open boundary condition ($L$ sites, $L-1$ bonds). Finally, we truncate the maximum bosonic number to $n_0 = 2$ at each site, which is justified due to low densities and strong repulsion among bosons.

## 2.2 Summary of results

In the XXZ-BH model, different orders compete with each other and, as a result, a plethora of quantum phases emerges. The key mechanism underlining the rich phase diagram is the bosonic Peierls mechanism, which we summarize in Sec. 3.1, driven by the the spin-dependent correlated tunneling term in the Hamiltonian (1), and accounting for long-range order in the system. Latter competes with different order patterns generated by the spin-spin interactions, giving rise to incommensurability effects. Let us first consider certain limiting cases.

For $J \ll \alpha$ the spins choose the uniform configuration $|\uparrow\uparrow\uparrow\uparrow \ldots\rangle$, with $M = L-1$, in order to maximize the bosonic tunneling, and as a result the system boils down to the standard Bose-Hubbard model. This occurs for any number of bosons $N$, which are then in a quasi-superfluid (qSF) phase except at integer fillings, where they form a Mott insulator (MI). The direct transition between the qSF and the MI phases occurs for $J \ll \alpha$ at a sufficiently high value of the chemical potential $\mu$, a parameter regime not considered in our work.

For $J \gg \alpha$, the spins are dominated by the XXZ Hamiltonian, and the ground state shows Néel order $|\uparrow\downarrow\uparrow\downarrow \ldots\rangle$. The effective Hamiltonian describing the bosons in this limit is a Bose-Hubbard model on a superlattice, where the tunneling elements are dimerized. Apart from the standard MI at integer fillings, this effective Hamiltonian has an insulating phase at $\rho = 1/2$ [36, 68, 69]. In this limit, the Néel-ordered spin state is doubly degenerate in the thermodynamic limit. This degeneracy is broken for finite chains and, depending on the spin configuration, the bosons will be either in a trivial or in a topological insulating state [36, 68].

At intermediate values of $J \sim \alpha$, the competition between these two terms gives rise to novel phases, which we summarize now. The emerging $(\mu/t, J/t)$ phase diagram is represented qualitatively in Fig. 1(b), showing two salient features: we find (1) phases with commensurate long-range order that have a non-trivial topological sector and (2) phases with incommensurate order that give rise to supersolidity.

(1) *Commensurate order and topology.* We find a parameter region where both the spins and the bosons present long-range commensurate order (CO) in the bonds for many values

of $N$ and $M$, that satisfy a one-to-one relation determined by the Peierls mechanism, as we will describe in the next section. Latter also characterizes the BOW pattern, which is different for every $(N, M)$ pair. These Peierls insulators extend the insulating phase at $J \gg \alpha$ for $N = L/2$ and $M = 0$ to other $(N, M)$ pairs, forming a Devil's staircase, where off-diagonal correlations fall off exponentially with the distance. This commensurate region is similar to the one previously found in [33] using a parallel magnetic field that competes with $\alpha$.

Moreover, we show that each of this bond-ordered bosonic Peierls insulators in the staircase is a symmetry-breaking topological insulator with fractionally-charged bosonic edge states, extending the results of [36, 37], where only the more stable steps were considered ($N = L/3$, $N = L/2$ and $N = 2L/3$). In particular, the Peierls transition occurs similarly for positive and negative values of $M$ and, while the former are topologically trivial, the latter present topological properties protected by certain symmetries (Fig. 1(c)).

(2) *Incommensurate order and supersolidity*. The inclusion of spin-spin interactions generates regions of incommensurate order (IO) around the CO region. There, both the spins and the bosons show long-range diagonal order, but $N$ and $M$ are now not in a one-to-one correspondence. This means that one can find certain ordered patterns characterized by the same $M$ for different values of $N$. Although this is expected for $J \gg \alpha$, with $M = 0$ and $N \neq L/2$, here we find this behavior also for $M \neq 0$, i.e., when the order in the spins is not of the Néel type. We call this mechanism *Peierls incommensurability*.

This mismatch with the one-to-one Peierls relation gives rise to a competition between the orders in the bosons and in the spins, resulting in off-diagonal algebraic correlations in the former, signaling superfluidity. This competition results in a SS phase, where off-diagonal quasi-long-range order and diagonal long-range order coexist. We refer to these phases as Peierls supersolids. We note that this new mechanism for supersolidity does not require the presence of long-range interactions. We also note that, due to the presence of superfluid nature in this IO region, the particle number $N$ changes smoothly with the variation of the chemical potential $\mu$ for a sufficiently long chain, making it a compressible phase.

# 3 Commensurate and Incommensurate orders

## 3.1 Bosonic Peierls mechanism

Before we describe the different phases of the model in detail, let us first revise the Peierls mechanism and its bosonic version, as it is central to understand the emergence of long-range order in the system. The Peierls mechanism was first introduce to explain the spontaneous symmetry breaking (SSB) of translational invariance on 1D fermionic chains coupled to phonon degrees of freedom, resulting in the so-called Peierls insulators [34]. This SSB leads to a larger unit cell in the system, opening two gaps in the fermionic part of the spectrum at certain momenta. The process is therefore favorable if the Fermi energy, given by the fermionic density, lies inside one of these gaps, as the energy of the occupied states would decrease. Peierls insulators present thus long-range order, which is characterized by a peak in the structure factor associated with the local order parameter. This peak is located at momentum $k_0$, which is smaller for larger unit cells, and the two gaps are opened at momenta $\pm k_0/2$. The process is thus energetically favorable when the following relation is satisfied,

$$1 - \frac{k_0}{\pi} = \left| 1 - 2\frac{|k_F|}{\pi} \right|, \tag{4}$$

where $k_F$ is the Fermi momentum, satisfying $|k_F| = \pi\rho$, with $\rho$ being the fermionic density. We note that, although in certain fermion-phonon systems, such as those described by the SSH

model, a Peierls instability always takes place at $T = 0$ [35], *quantum* Peierls transitions can also take place in the ground state between ordered and disordered phases [33, 36].

In the XXZ-BH model (3), for $U/t \to \infty$ we can map the hardcore bosons to spinless fermions, and the system presents a well defined Fermi surface. Although for a finite value of $U/t$ this is not possible, a bosonic Peierls transition can still take place if $U/t$ is sufficiently large, as shown in [33, 36]. Even if $k_F$ is not defined, we can still check that the Peierls relation is satisfied using the bosonic density $\rho$. Since here the order develops in the bonds, we can characterized it by measuring the bosonic tunneling $\hat{B}_i = \hat{b}_i^\dagger \hat{b}_{i+1} + \text{H.c.}$, and its corresponding structure factor,

$$S_B(k) = \frac{1}{L^2} \sum_{j_1, j_2} e^{i(j_1 - j_2)k} \left( \langle \hat{B}_{j_1} \hat{B}_{j_2} \rangle - \langle \hat{B}_{j_1} \rangle \langle \hat{B}_{j_2} \rangle \right). \tag{5}$$

Moreover, the same order develops for the spins in the bonds, which can be characterized using the spin structure factor,

$$S_\sigma(k) = \frac{1}{L^2} \sum_{j_1, j_2} e^{i(j_1 - j_2)k} \langle (\hat{\sigma}_{j_1}^z - m)(\hat{\sigma}_{j_2}^z - m) \rangle, \tag{6}$$

where $m = M/L$ is the density of magnetization. Here both $S_B$ and $S_\sigma$ develop a peak at the same momentum $k_0$. The Peierls relation then takes the form $|1 - k_0 \pi| = |1 - 2\rho|$, which generalizes Eq. (4) to bosonic matter. We can detect if the latter is fulfilled or not by measuring the incommensurability parameter

$$\mathcal{I}(\rho, m) = |1 - 2\rho| - \left( 1 - \frac{k_0}{\pi} \right), \tag{7}$$

which is zero when the Peierls relation is satisfied, and non-zero otherwise. We refer to the first case as commensurate order (CO) and to the second case as incommensurate order (IO).

Since in our model the total magnetization $M$ is conserved and can only take integer values, $m$ is restricted to rational numbers between $-1$ and $1$. In this situation, $k_0$ is completely characterized by $m$ through the relation $k_0/\pi = 1 - |m|$, and the incommensurability parameter can be written as

$$\mathcal{I}(\rho, m) = |1 - 2\rho| - |m|, \tag{8}$$

which reduces to a relation between two conserved quantities associated with the two global $U(1)$ symmetries of the model. Notice that the Peierls relation does not depend on the sign of the magnetization. We shall see how different signs lead to different topological properties.

Finally, we note that for a finite chain with $L$ sites and $L - 1$ bonds, the density of magnetization is to be redefined as $m = M/(L - 1)$ and the relation is to be modified due to boundary conditions as

$$\mathcal{I}(\rho, m) = |1 - 2\rho| - |m| + (1 + |m|)/L. \tag{9}$$

## 3.2 Staircases of Peierls insulators

As introduced in the previous subsection, in the regions with CO the discrete translational symmetry gets spontaneously broken by enlarging the unit cell, which results in a finite gap in the spectrum (i.e., Peierls insulator) and the peak in the spin structure factor $S_\sigma(k)$ is located at $k_0 \pi = 1 - |m|$ for infinite systems. In that case, rational values of the density of magnetization $m$ and the particle density $\rho$ can be written as

$$(1 - |m|)/2 = \frac{p_1}{q_1}, \quad \rho = \frac{p_2}{q_2}, \tag{10}$$

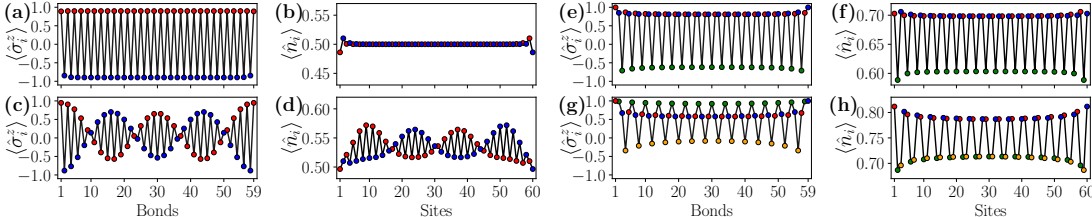

Figure 2: **Bosonic Peierls insulators:** We show the real-space patterns of spins ($\langle \hat{\sigma}_i^z \rangle$) and number occupation ($\langle \hat{n}_i \rangle$) for systems with densities $\rho = 1/2$ **(a)**-**(b)**, $8/15$ **(c)**-**(d)**, $2/3$ **(e)**-**(f)**, and $3/4$ **(g)**-**(h)** in the Peierls insulators for a chain of size $L = 60$. Here we have set $J/t = 0.6, 0.4, 0.32$, and $0.28$ corresponding to densities $\rho = 1/2, 8/15, 2/3$, and $3/4$ respectively, and densities are fixed by using $U(1)$ symmetric MPS that preserves particle number. Different colors represent different element of the unit cell.

and the Peierls relation (Eq. (8)) is satisfied if $p_1/q_1 = p_2/q_2$. For finite systems with open boundaries, the relation again has to be modified as

$$\frac{1}{2}\left(1 - |m| + \frac{(1+|m|)}{L}\right) = \frac{p_1}{q_1}, \quad \rho = \frac{p_2}{q_2}. \tag{11}$$

By first decomposing the lattice translational symmetry into a translation of unit cells of length $q_1 = q_2 \equiv q$ and a $\mathbb{Z}_q$ symmetry describing translations modulo $q$ within the unit cells, one can understand the Peierls transition as a SSB of the $\mathbb{Z}_q$ symmetry. Since rational numbers are dense and $q$ can take any integer value between 2 to $L-1$, in the thermodynamic limit we can have infinitely many Peierls insulators, each of them possessing diagonal long-range order and a unit cell of size $q$, forming a Devil's staircase. It is important to note that not every step is equally stable to quantum or thermal fluctuations, since the insulating gap is different for each of them, and only certain Peierls insulators might appear in the phase diagram. Here we show,

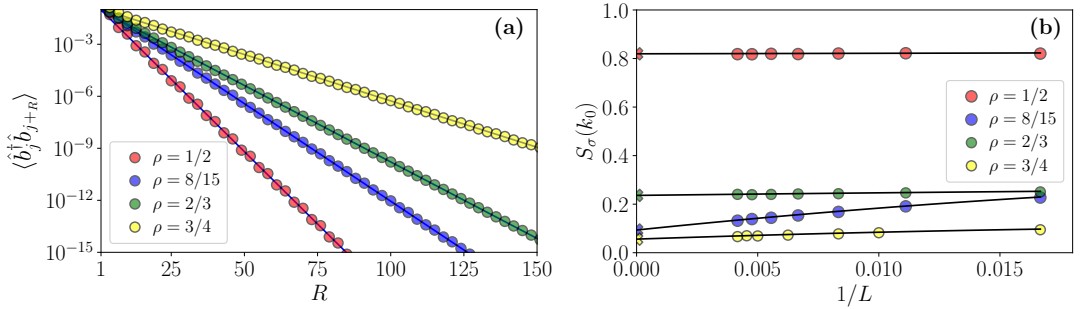

Figure 3: **Correlations in the Peierls insulators: (a)** We plot the off-diagonal correlations $\langle \hat{b}_j^\dagger \hat{b}_{j+R} \rangle$ in the Peierls insulators with densities $\rho = 1/2, 8/15, 2/3$, and $3/4$ as a function of the distance $R$ for a system of size $L = 180$. Clearly, the correlations $\langle \hat{b}_j^\dagger \hat{b}_{j+R} \rangle$ show exponential decays (solid black lines) with respect to $R$ as expected for insulating phases. **(b)** The variations in the peak of the structure factor $S_\sigma(k)$ with system-sizes $L \in [60, 240]$ in the Peierls insulators having same densities as in **(a)**. We extract non-zero values of the peak $S_\sigma(k_0)$ in the thermodynamic limit ($L \to \infty$) by extrapolating the finite-size data using second-order polynomials in $1/L$ as shown by solid black lines. The error-bars in the fits are smaller than the symbol sizes.

however, that the complete staircase appears for the considered parameter region. Fig. 2 gives four examples of Peierls insulators for a chain of size $L = 60$, with bosonic densities $\rho = 1/2$, 8/15, 2/3 and 3/4, resulting in $\mathbb{Z}_2$, $\mathbb{Z}_{15}$, $\mathbb{Z}_3$ and $\mathbb{Z}_4$ long-range order, respectively. Although not depicted in the figure, the bosonic bonds $\langle \hat{B}_i \rangle$ present the same ordered structure as the spins.

To confirm that these phases are indeed insulators and do not fit the descriptions of Luttinger liquid theory [70], we analyze the decay in the off-diagonal correlations $\langle \hat{b}_j^\dagger \hat{b}_{j+R} \rangle$ with the distance $R$. Figure 3(a) shows that these off-diagonal correlations decay exponentially with $R$ following $\langle \hat{b}_j^\dagger \hat{b}_{j+R} \rangle \propto \exp(-R/\xi)$ confirming the insulating nature of these phases. Furthermore, in Fig. 3(b) we study the variations in the peak of the structure factor $S_\sigma(k)$ with the size of the system in these Peierls insulators, where the position of the peaks are determined by $k_0/\pi = 1 - |1 - 2\rho|$ corresponding to the commensurability condition $\mathcal{I}(\rho, m) = 0$. By extrapolating the finite-size data to the thermodynamic limit using second-order polynomials in $1/L$, we obtain non-vanishing values of $\lim_{L\to\infty} S_\sigma(k_0)$ that testify the existence of diagonal long-range order in these insulating phases.

The Devil's staircase pattern of the XXZ-BH model is shown in Fig. 4, where we show variations in the peak of the structure factor $S_\sigma(k)$, the incommensurability parameter $\mathcal{I}$, the density $\rho$ and the magnetization density $m$ with $\mu/t$ ($J/t$) for a fixed value of $J/t$ ($\mu/t$) in the ground state of a finite chain with $L = 60$ sites. As the chemical potential $\mu$ or the XXZ interaction strength $J$ is changed, we observe a staircase of different plateaus in the values of these quantities. In the CO region, i.e., the region with $\mathcal{I} = 0$ in Fig. 4, at every step $S_\sigma(k)$ presents a peak with a non-zero height at $k_0 = (1 - |m|)/\pi = \frac{2p_1}{q\pi}$, signaling the bond order. We have, therefore, a staircase of Peierls insulators where each step is a BOW with a broken $\mathbb{Z}_q$ symmetry. We note that the transitions between different steps correspond to first order phase transitions, as signaled by the discontinuous jumps in the structure factor (Fig. 4).

### 3.3 Topological Peierls insulators

We show now that, apart from the long-range order, every Peierls insulator in the staircase possesses also non-trivial topological properties. We first note that, as introduced in Sec. 3.1, the Peierls relation depends only on the absolute value of the magnetization, but not on its sign (Eq. (8)). The latter can be controlled by adding an external field to the Hamiltonian, $\hat{H}_z = h_z \sum_i \hat{\sigma}_i^z$. While both positive and negative signs of $M$ lead to the same type of long-range order, it turns out that the resulting Peierls insulators present different topological properties. In particular, while positive signs lead to trivial insulators, negative signs lead to topological Peierls insulators with localized states at the edges of the chain.

In Fig. 5 we present two examples for different values of $\rho$ (1/2 (left column) and 2/3 (right column)) and $m$ (negative). In both cases, the bosonic occupation present peaks at the edges, associated to fractional bosonic edge states. In this particular case, the right edge is occupied and the left one is empty, corresponding to $+1/2$ and $-1/2$ boson, respectively. To verify these fractional excitations, we consider site-resolved density deviation, $\delta \langle \hat{n}_i \rangle = \langle \hat{n}_i \rangle - \rho$ and the corresponding integrated density deviation

$$\delta N_i = \sum_{j \leq i} \delta \langle \hat{n}_j \rangle . \tag{12}$$

In the particular examples, the integrated density deviation (as shown in Figs. 5(e) and (f)) becomes $1/2$ as soon as the left edge is crossed (corresponding to $+1/2$ boson in the left edge) and remains fixed at $1/2$ in the bulk (modulo the deviation coming from density pattern in the case of $\rho = 2/3$). On the other hand, it suddenly goes to zero in the right edge signaling a $-1/2$ particle excitation.

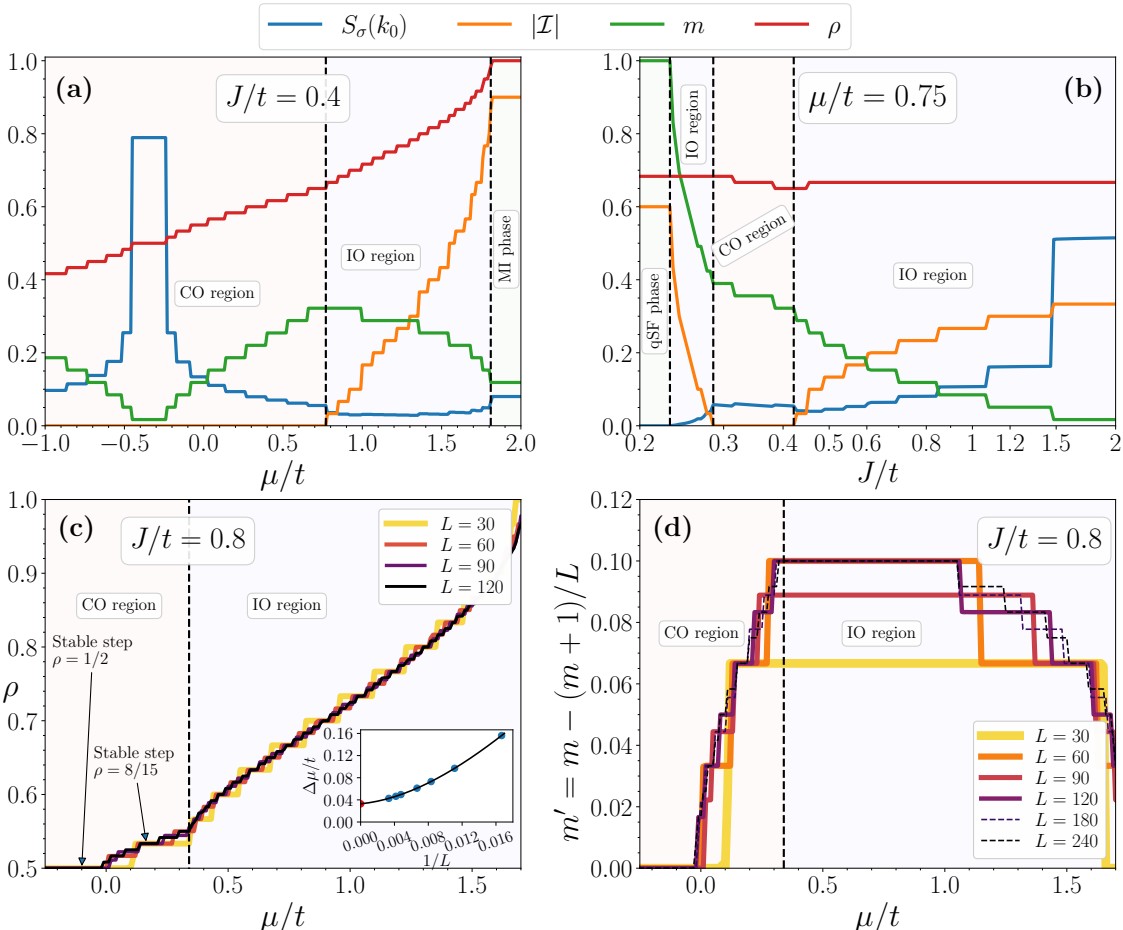

Figure 4: **Devil's staircases: (a)-(b)** we show the height of the structure factor peak $S_\sigma(k_0)$, the incommensurability parameter $\mathcal{I}$, the bosonic density $\rho$ and the magnetization density $m$ for a chain of $L = 60$ sites. These quantities are shown as a function of $\mu/t$ **(a)** and $J/t$ **(b)** for fixed values of $J/t = 0.4$ (lower horizontal dotted line in Fig. 1**(b)**) and $\mu/t = 0.75$ (vertical dotted line in Fig. 1**(b)**), respectively. The Devil's staircase structure become apparent in the commensurate region. **(c)** Bosonic density $\rho$ as a function of the chemical potential $\mu/t$ for fixed $J/t = 0.8$ (upper horizontal dotted line in Fig. 1**(b)**) and for different system-sizes $L = 30, 60, 90, 120$. The steps in terms of $\rho$ are stable in the commensurate region. As pointed out in the text, some steps (corresponding to lower integer values of $q$ in $\rho = p/q$) are more stable, as seen in the figure for $\rho = 1/2$ and $8/15$. In the inset of **(c)**, we show the stability of the $\rho = 8/15$ step in the thermodynamic limit by extrapolating the step-size in $\mu/t$. On the contrary, the steps (in terms of $\rho$ vs. $\mu/t$) in the incommensurate region are not stable with increasing system-size (and in the thermodynamic limit) showing that the incommensurate region is compressible. **(d)** The modified magnetization density $m' = m - (m + 1)/L$ (needed to compare different system-sizes with open boundary condition having $L-1$ spin degrees of freedom) as a function of the chemical potential $\mu/t$ for fixed $J/t = 0.8$. The plateaux seen in $m'$ verify the stability of the steps in terms of the spin degrees of freedom in the IO region.

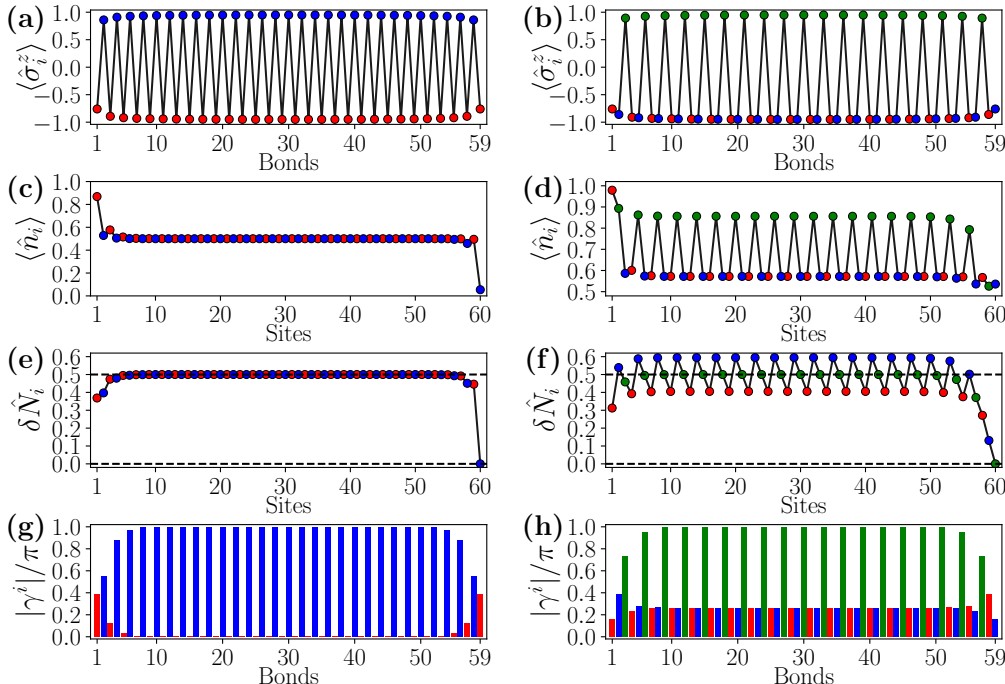

Figure 5: **Symmetry-protected topological edge states:** Peierls insulators present non-trivial topological properties for $m < 0$. In the figure, for a chain of size $L = 60$ we show the real-space patterns of the spins ($\langle \hat{\sigma}_i^z \rangle$) (**(a)** and **(b)**) and the number occupation ($\langle \hat{n}_i \rangle$) (**(c)** and **(d)**) located at each bond and site, respectively. The left column corresponds to the density $\rho = 1/2$ and the right column $\rho = 2/3$. In both cases, we can observe peaks in the occupation associated to occupied (left) and empty (right) edge states associated with bosons with a fractional number ($\pm 1/2$). This contrasts with the corresponding patterns of the trivial insulators depicted in Fig. 2, where the edge states are absent. The fractional excitations are further illustrated in **(e)** and **(f)** by means of integrated density deviation $\delta \hat{N}_i$ (see text). **(g)** and **(h)** The local many-body Berry phases $\gamma^i$ provides the bulk-boundary correspondence and serves as local order parameters in the topological insulators. Specifically, $\gamma^i$ are quantized to 0 or to $\pi$ in the bulk for the half-filling case, while for $\rho = 2/3$ it is equal to $\pi$ for the bonds that respect inversion symmetry.

These topological Peierls insulators are thus examples of symmetry-breaking topological phases [71], showing both long-range order and non-trivial topological effects. The latter are protected here by an inversion symmetry with respect to the central bond of the chain. Notice the this symmetry is preserved even if the $\mathbb{Z}_q$ symmetry is spontaneously broken. In order to show a convincing argument in favor of the topological characters of the studied phases, we consider *local* many-body Berry phase [72]. Following the prescription of [36], many-body Berry phase is defined as

$$\gamma^i = \text{Arg} \prod_{n=0}^{K-1} \langle \psi_{\theta_{n+1}}^i | \psi_{\theta_n}^i \rangle , \tag{13}$$

where $|\psi_{\theta_n}^i\rangle$'s are the ground states of the Hamiltonian (3) with a local twist $t \to t e^{i\theta_n}$ at $i^{th}$ bond with $\theta_0, \theta_1, ..., \theta_K = \theta_0$ on a loop in $[0, 2\pi]$. In case of density $\rho = 1/2$ (Fig. 5**(g)**), the local Berry phases in the bulk are quantized to 0 or $\pi$ for weak and strong bonds respectively. On the other hand, the Berry phases are again quantized to $\pi$ for the bonds that respect the

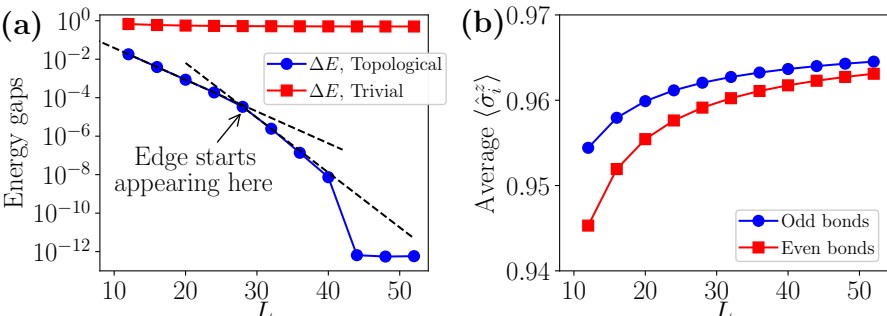

Figure 6: **Topological degeneracy:** **(a)** We plot the energy gap between the first two eigenstates with system size $L$ for both topological (blue circles) and trivial (red squares) Peierls insulators for density $\rho = 1/2$. The energy gap in the topological case reaches the precision of the DMRG calculation for $L = 44$. **(b)** The variation of bond-averaged $\langle \hat{\sigma}_i^z \rangle$ with system size $L$, for even and odd bond. This shows that the tunneling elements become more dimerized when the system size increases, causing a reduction of the correlation length.

inversion symmetry in the scenario of density $\rho = 2/3$ (Fig. 5**(h)**).

In Fig. 5 we have represented one of the two possible edge states configurations, where the other one (not shown) corresponds to the right edge being occupied and the left edge being empty. These two states are degenerate in the thermodynamic limit, and this degeneracy is yet another hallmark of the non-trivial topology where the energy gap closes exponentially fast with increasing system size [73]. This does not occur for the corresponding trivial Peierls insulators for positive values of $m$ (Fig. 2**(a)** and **(b)**). For finite systems, these two states are not completely degenerate, but the energy difference between them decreases with the system size. This gap closing is usually exponential in the system size, signaling an underlying topological property. However, in our case we find that the gap closing is faster than exponential (Fig. 6**(a)**). The reason for this behavior is the following. For small system sizes, when the correlation length is larger than the system itself, the edge states hybridize among themselves and fractional particle-hole excitations do not appear on real-space profiles. When the system size becomes larger than the correlation length, edge states starts appearing (e.g., $L = 28$ in case of Fig. 6**(a)**), and the gap closing accelerates. On the other hand, the tunneling elements get more dimerized with $L$, as shown by the average spin densities in Fig. 6**(b)**. As the result, the correlation itself slowly reduces causing a faster-than-exponential gap-closing.

## 3.4 Peierls supersolids

Apart from the CO region described in the previous sections, we find regions in the phase diagram characterized also by a long-range order but, contrary to the former, the Peierls relation is now not fulfilled. This incommensurate order is distinguished by a non-zero value of the incommensurability parameter (8). Figure 4 shows how, next to the staircase of Peierls insulators in the CO region, there is another region where both $S_\sigma(k_0)$ and $\mathcal{I}$ are non-zero simultaneously. In this case, $\rho$ and $m$ are not in the one-to-one correspondence given by the Peierls relation. In particular, Fig. 4**(b)** shows how, for the same density $\rho$, one finds different values of $m$ as $J/t$ is increased, and all of the corresponding states present long-range order.

As opposed to the states with commensurate order, in the IO region the long-range ordered pattern for the spins (6) and the bosonic bonds (5) does not coincide. In particular, the periodicity of the spin subsystem, this is, the size of the new unit cell after SSB of translational

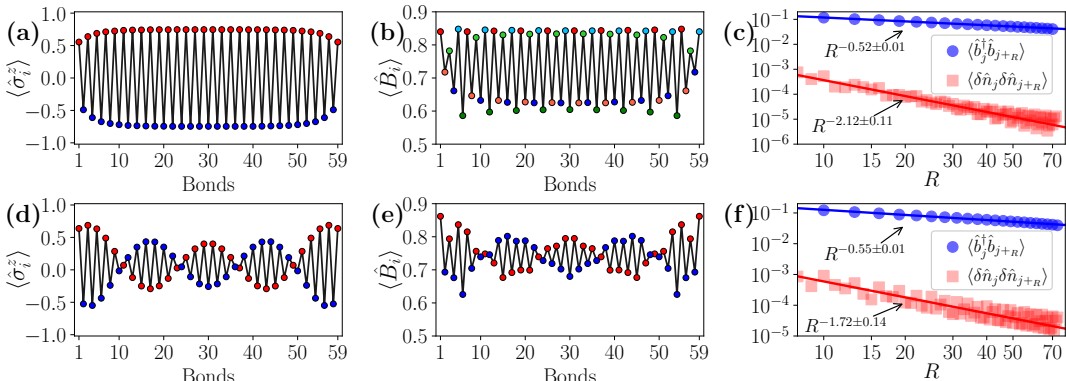

Figure 7: **Peierls incommensurability and supersolids:** Real-space ordered patterns and superfluid correlations of the Peierls supersolids for density $\rho = 2/3$ and incommensurability parameter $\mathcal{I} = 1/3$ (top row) and $4/15$ (bottom row). Here we have set $J/t = 2$ and $J/t = 0.9$ respectively, and the density of bosons is fixed by using $U(1)$ symmetric MPS that conserves particle number. **(a)** and **(d)** show the spin pattern given by $\langle \hat{\sigma}_i^z \rangle$ for a system of size $L = 60$, while **(b)** and **(e)** show the bosonic tunneling $\langle \hat{B}_i \rangle$. Notice that the periodicity in the ordered structure is different for these two quantities. **(c)** and **(f)** show the algebraic decay of the long-range correlations $\langle \hat{b}_j^\dagger \hat{b}_{j+R} \rangle$ and $\langle \delta \hat{n}_j \delta \hat{n}_{j+R} \rangle$ with the distance $R$ for a system of $L = 120$ sites.

invariance, is still characterized by $q_1$, which is given by $m$ (10). However, the periodicity in the structure of the bosonic bonds is now not given by $q_2$ (10), since the Peierls relation is not fulfilled, but neither by $q_1$. The new order that develops in the bonds is instead characterized by the least common multiple $q = \text{lcm}(q_1, q_2)$. It is therefore the result of the competition between the order in the spins and the order given by the Peierls mechanism. Fig. 7 depicts the spatial patterns corresponding to two different steps in the IO, for the same density $\rho = 2/3$ and different values of $m$, and thus different values of the incommensurability parameter, $\mathcal{I} = 1/3$ and $4/15$. The lattice periodicities are 6 and 15, respectively.

Since bond order develops for each step in the IO region, each one corresponds to a BOW phase. However, contrary to the steps in the CO regions, these are not insulators. In this case, each step of the staircase in the IO region corresponds to a Peierls supersolid, where the long-range diagonal order coexists with off-diagonal algebraically-decaying correlations. Such off-diagonal quasi-long-range order arises as the orders in the spins and in the bosons are now in competition with each other, and the extra particles over the Peierls commensurate relation $N - N^*$, with $\mathcal{I}(N^*, M) = 0$, can move freely forming a gapless fluid. Due to these superfluidic character in the IO region, the system becomes compressible, where the density plateaux in the variation of the chemical potential $\mu$ (Fig. 4**(c)**) reduce to points with $\frac{\partial \rho}{\partial \mu} \neq 0$ in the thermodynamic limit. However, the staircase structure remains stable when the plateaux are considered in terms of other suitable order parameters like the magnetization density $m$ or the structure factor $S_\sigma(k_0)$ (see Fig. 4**(d)**).

In Fig. 7**(c)** and Fig. 7**(f)** we show both the off-diagonal $\langle \hat{b}_j^\dagger \hat{b}_{j+R} \rangle$ and density-density correlations $\langle \delta \hat{n}_j \delta \hat{n}_{j+R} \rangle$ with $\delta \hat{n}_j = \hat{n}_j - \langle \hat{n}_j \rangle$. Both decay algebraically with the distance $R$, i.e.,

$$
\langle \hat{b}_j^\dagger \hat{b}_{j+R} \rangle = c_1 R^{-\nu_1},
$$
$$
\langle \delta \hat{n}_j \delta \hat{n}_{j+R} \rangle = c_2 R^{-\nu_2}, \tag{14}
$$

confirming the gapless superfluid character of the state. Moreover, we find that the exponents

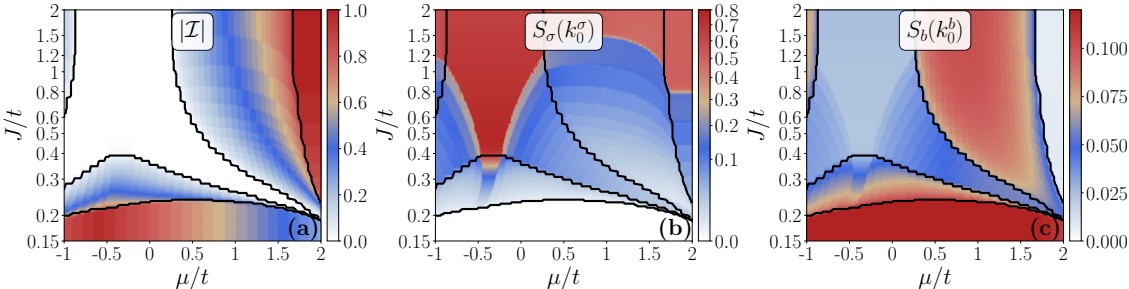

Figure 8: **Phase diagram:** We represent three different quantities as a function of $\mu/t$ and $J/t$ for a chain of size $L = 60$, which allow us to completely characterize the different regions of the XXZ-Bose-Hubbard model. These are the incommensurability parameter $\mathcal{I}$ **(a)**, the maximum of the spin structure factor $S_\sigma(k_0^\sigma)$ **(b)** and the maximum value of the off-diagonal structure factor $S_b(k_0^b)$ **(c)**. The black lines differentiates the different phases regions, depicted also in Fig. 1**(b)**. The staircase structures can be appreciated inside each region by means of the spin structure factor.

follow the relation $\nu_2 \approx 1/\nu_1$, which tells us that the low energy description of these SS phases are well described by Luttinger liquid theory [70]. These results confirm the presence of a staircase of Peierls supersolids in the XXZ-Bose-Hubbard model, showing how cold atoms can not only simulate similar phenomena as those found in solid state, such as Peierls insulators, but to extend them to generate novel phases of matter.

The phase diagram of the model is summarized in Figure 8, where we represent both the incommensurability parameter and the spin structure factor, which helps to distinguish the commensurate from the incommensurate long-range order. The superfluid character of the latter is identified using the off-diagonal structure factor in bosons (Fig. 8**(c)**), defined as

$$S_b(k) = \frac{1}{L^2} \sum_{j_1 \neq j_2} e^{i(j_1 - j_2)k} \langle \hat{b}_{j_1}^\dagger \hat{b}_{j_2} \rangle. \tag{15}$$

For insulating phases $S_b(k)$ is vanishing in the thermodynamic limit, whereas in the phases with qSF order a peak in the structure factor at a momentum $k_0^b$ attains a finite value. Clearly, $S_b(k_0^b)$ possesses high values in the usual SF phase and in the staircase of incommensurate Peierls SS, while it is visibly smaller (non-zero due to finite size effects) in the Peierls insulator and in the Mott phase. In the following table, we summarize the different regions that appear in the phase diagram and the order parameters used to distinguished them.

|  | SF | MI | CO | IO |
|---|---|---|---|---|
| $\mathcal{I}$ | $\neq 0$ | $\neq 0$ | $= 0$ | $\neq 0$ |
| $S_\sigma(k_0^\sigma)$ | $= 0$ | $-$ | $\neq 0$ | $\neq 0$ |
| $S_b(k_0^b)$ | $\neq 0$ | $= 0$ | $= 0$ | $\neq 0$ |

# 4 Atomic quantum simulation

We now show how the XXZ-BH model (3) provides an effective description for a bosonic mixture of ultracold atoms in an optical lattice. Let us consider, in particular, two atomic species, a and b, trapped by two different optical lattices, where the lattice spacing of one lattice is twice as large as the spacing in the other lattice, such that, for the latter, both coincide for

every second minimum (see Fig. 9**(a)**). For the a atoms trapped by the lattice with double spacing we consider two internal states, that we denote ↑ and ↓. In such configuration, the atomic system is described by the following Hamiltonian,

$$\hat{H} = \hat{H}_{\text{b}} + \hat{H}_{\text{a}} + \hat{H}_{\text{ba}}, \tag{16}$$

where

$$\hat{H}_{\text{b}} = -t_{\text{b}} \sum_i \left(\hat{b}_i^\dagger \hat{b}_{i+1} + \text{H.c.}\right) + \frac{U^{\text{b}}}{2} \sum_i \hat{n}_i^{\text{b}}(\hat{n}_i^{\text{b}} - 1) \tag{17}$$

and

$$\hat{H}_{\text{a}} = -t_{\text{a}} \sum_{i,\sigma} \left(\hat{a}_{2i,\sigma}^\dagger \hat{a}_{2i+2,\sigma} + \text{H.c.}\right) + U_{\uparrow\downarrow}^{\text{a}} \sum_i \hat{n}_{2i,\uparrow}^{\text{a}} \hat{n}_{2i,\downarrow}^{\text{a}} + \frac{1}{2} \sum_{i,\sigma} U_\sigma^{\text{a}} \hat{n}_{2i,\sigma}^{\text{a}}(\hat{n}_{2i,\sigma}^{\text{a}} - 1) \tag{18}$$

are the standard BH Hamiltonians describing bosonic atoms in the presence of a nearest-neighbor tunneling $t_{\text{a/b}}$ and on-site Hubbard interactions $U^{\text{b}}$, $U_\uparrow^{\text{a}}$, $U_\downarrow^{\text{a}}$ and $U_{\uparrow\downarrow}^{\text{a}}$. Finally, the interspecies interactions on every second site are given by

$$\hat{H}_{\text{ab}} = \sum_{i,\sigma} \hat{n}_{2i}^{\text{b}} \left(U_\uparrow^{\text{ab}} \hat{n}_{2i,\uparrow}^{\text{a}} + U_\downarrow^{\text{ab}} \hat{n}_{2i,\downarrow}^{\text{a}}\right), \tag{19}$$

where $U_\uparrow^{\text{ba}}$ and $U_\uparrow^{\text{ba}}$ are the corresponding interspecies interaction parameters.

We now consider the limit $t_{\text{a}} \ll U_\uparrow^{\text{a}}, U_\downarrow^{\text{a}}, U_{\uparrow\downarrow}^{\text{a}}$. If the system is initialized with one atom of the a species on every site, the resulting state will be a MI. In this limit, we can work in a subspace where the number of atoms is fixed on every site, $\sum_{i,\sigma} \hat{n}_{2i,\sigma}^{\text{a}} = 1$, becoming, effectively, a conserved quantity. The b atoms are initialized in such a way that only odd sites are occupied. If we now work in the limit $t_{\text{b}} \ll U_\uparrow^{\text{ab}}, U_\downarrow^{\text{ab}}$, the tunneling of the b atoms between even and odd sites is inhibited. However, they can tunnel two sites apart through second order processes (Fig. 9**(b)**). This effective tunneling depends on the state of the a atom on the even sites, and it is described by the Hamiltonian

$$\hat{H}_{\text{eff}} = -\sum_{i,\sigma} \frac{t_{\text{b}}^2}{U_{\text{ba}}^\sigma} \left(\hat{b}_{2i-1}^\dagger \hat{n}_{2i,\sigma}^{\text{a}} \hat{b}_{2i+1} + \text{H.c.}\right). \tag{20}$$

Since the total number of a atoms on each site is conserved, we can described them using spin variables,

$$\hat{\sigma}_{2i}^z = \hat{n}_{2i,\uparrow}^{\text{a}} - \hat{n}_{2i,\downarrow}^{\text{a}}, \quad \hat{\sigma}_{2i}^x = \hat{a}_{2i,\uparrow}^\dagger \hat{a}_{2i,\downarrow} + \text{H.c.}, \tag{21}$$

where $\hat{\sigma}_{2i}^x$ and $\hat{\sigma}_{2i}^z$ are the standard Pauli matrices. Using the spin description, the effective correlated tunneling terms (20) can be written as

$$\hat{H}_{\text{eff}} = -\sum_i \left[\hat{b}_{2i-1}^\dagger \left(t + \alpha \hat{\sigma}_{2i}^z\right) \hat{b}_{2i+1} + \text{H.c.}\right], \tag{22}$$

where $t = t_{\text{b}}^2/2 \left(1/U_{\text{ba}}^\uparrow + 1/U_{\text{ba}}^\downarrow\right)$ and $\alpha = t_{\text{b}}^2/2 \left(1/U_{\text{ba}}^\downarrow - 1/U_{\text{ba}}^\uparrow\right)$. If we now rename the sites $2i - 1 \to i$, $2i \to i$, Eq. (22) becomes precisely the correlated-tunneling term that appears in the $\mathbb{Z}_2$BH model (1). Notice, in particular, that the ratio between the direct tunneling and the spin-dependent tunneling can be tuned at will by modifying the difference between the two interspecies scattering channels, $\alpha/t = \left(U_{\text{ba}}^\downarrow - U_{\text{ba}}^\uparrow\right)/\left(U_{\text{ba}}^\downarrow + U_{\text{ba}}^\uparrow\right)$, which can be done using a Feshbach resonance. In the following, we drop the b index to simplify the notation.

The XXZ interactions can be readily introduced in the implementation described above by reducing the lattice depth for the a species, which controls the ratio between the tunneling and the intraspecies Hubbard interactions. Second-order tunneling processes in the Mott insulator

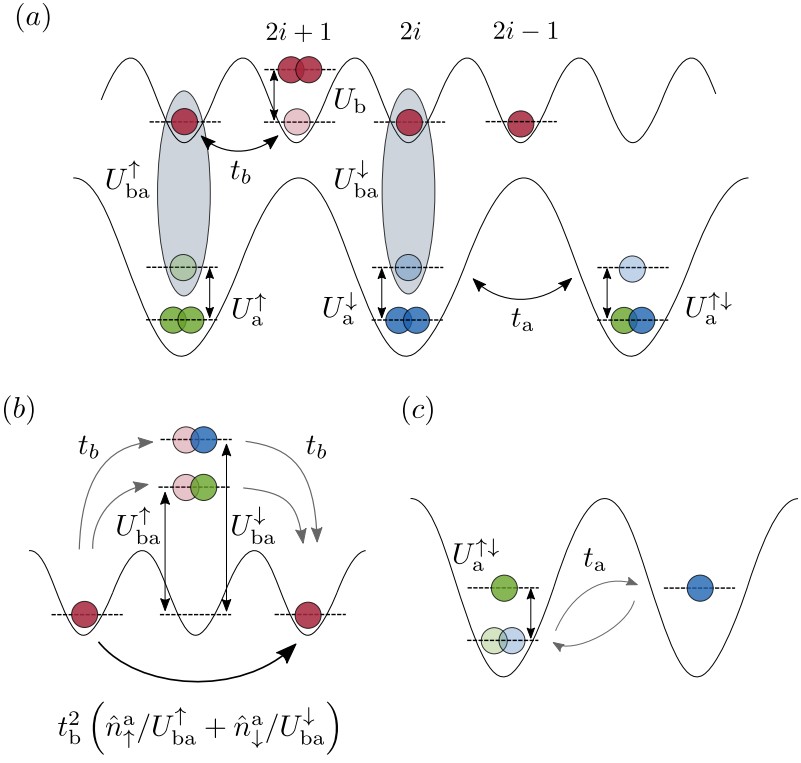

Figure 9: **Cold-atom quantum simulator:** **(a)** Lattice structure for the proposed experimental scheme to implement the XXZ-BH model, showing two optical lattices with different periodicity. One lattice with a shorter wavelength traps atoms of the species b (red), that can tunnel to nearest-neighbor minima with a coefficient $t_b$ and interact on-site with a Hubbard coefficient $U_b$. The second lattice has a wavelength twice as large, and traps atoms of another species a. The latter present two internal states, representing spin up (green) and down (blue) states. Both tunnel with a coefficient $t_a$ and interact with coefficients $U_a^\uparrow$, $U_a^\downarrow$ and $U_a^{\uparrow\downarrow}$. Finally, both species interact with coefficients $U_{ba}^\uparrow$ and $U_{ba}^\downarrow$. **(b)** For the right choice of initial conditions (see main text), the direct tunneling of the b bosons is suppressed. They can still tunnel two sites apart via second-order processes through intermediate higher-energy states, resulting in a correlated tunneling mediated by the spin state of the a atoms. **(c)** Spin-spin exchange interactions also appear through second-order processes involving a back and forth tunneling of the a atoms.

(MI) give rise to Heisenberg-like interactions between nearest-neighbor spins [12], where the coefficients in Eq. (2) depend on the lattice depth and the different atomic scattering lengths, $J = -4t_A^2/U_A^{\uparrow\downarrow}$ and $J\Delta = 4t_A^2\left(1/U_A^{\uparrow\downarrow} - 1/U_A^\uparrow - 1/U_A^\downarrow\right)$. In particular, the anisotropy $\Delta$ can be tuned using a Feshbach resonance, such that $U_A^{\uparrow\downarrow}, U_A^\uparrow, U_A^\downarrow < 0$ and the interactions are antiferromagnetic, as shown in a recent experiment with bosonic atoms [31]. Combining all these ingredients together we obtain all the necessary terms to simulate the XXZ-BH model (3). We note, however, that extra terms can appear in the effective description. In particular density-spin interactions of the form $(\hat{n}_i + \hat{n}_{i+1})\hat{\sigma}_i^z$ also appear as second-order processes. A more careful analysis should be performed to analyze the importance of these terms, as well as possible ways to suppress them if it is necessary.

# 5 Conclusions and outlook

In this work, we investigated the XXZ-Bose-Hubbard model, where a spin-dependent correlated tunneling coexists with spin-spin XXZ interactions, and show how it emerges as the effective description of a bosonic mixture of ultracold atom in optical lattices. First, we introduced the notion of bosonic Peierls transitions, where, similarly to fermion-phonon systems in solid-state physics, strongly-correlated bosonic matter can break spontaneously the lattice translational invariance, giving rise to bosonic Peierls insulators with the bond long-range order. Even in the absence of long-range interactions, the phase diagram of the model presents a Devil's staircase structure of Peierls insulators where, on each step, both the bosonic density and the spin magnetization are in a one-to-one correspondence, and characterize different ordered patterns through the Peierls relation. Moreover, we show how each step presents non-trivial topological properties, such as fractionalized edge states and a topological degeneracy.

Moreover, we uncovered another region of BOW phases which, instead of being insulators, present superfluid correlations. These Peierls supersolids arise due to a competition between two types of order in the system, the magnetic order in the spin and the bond order in the bosons given by the Peierls relation. These supersolid phases driven by such Peierls incommensurability are novel phase of matter that shows the possibilities that synthetic atomic platforms offer to explore strongly-correlated phenomena that, although inspired by solid-state physics, go beyond those found in natural materials.

# Acknowledgements

**Funding information** The support by National Science Centre (Poland) under project Unisono 2017/25/Z/ST2/03029 (T.C. and J.Z.) within QuantERA QTFLAG is acknowledged. The continuous support of PL-Grid Infrastructure made the reported calculations possible. L.T. acknowledges support from the Ramón y Cajal program RYC-2016-20594, the "Plan Nacional Generación de Conocimiento" PGC2018-095862-B-C22 and the State Agency for Research of the Spanish Ministry of Science and Innovation through the "Unit of Excellence María de Maeztu 2020-2023" award to the Institute of Cosmos Sciences (CEX2019-000918-M). D.G.-C. and M.L. acknowledge support from the European Union Horizon 2020 research and innovation programme under the Marie Skłodowska-Curie grant agreement No 665884, ERC AdG NOQIA, Spanish Ministry of Economy and Competitiveness ("Severo Ochoa" program for Centres of Excellence in R&D (CEX2019-000910-S), Plan National FIDEUA PID2019-106901GB-I00/10.13039 / 501100011033, FPI), Fundació Privada Cellex, Fundació Mir-Puig, and from Generalitat de Catalunya (AGAUR Grant No. 2017 SGR 1341, CERCA program, QuantumCAT U16-011424, co-funded by ERDF Operational Program of Catalonia 2014-2020), MINECO-EU QUANTERA MAQS (funded by State Research Agency (AEI) PCI2019- 111828-2 / 10.13039/501100011033), EU Horizon 2020 FET-OPEN OPTOLogic (Grant No 899794), and the National Science Centre, Poland-Symfonia Grant No. 2016/20/W/ST4/00314.

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
