# Peer review of "Devil's staircase of topological Peierls insulators and Peierls supersolids"

_SciPost Physics, doi:SciPost Phys. 12, 076 (2022)_

## Round 4 · Referee Report · Anonymous · 2021-12-13

Strengths
see previous report.
Weaknesses
I still feel the manuscript has a difficulty in providing an understanding for the presence of a *staircase* of ordered phases. For example the step at rho=8/15 is prominently discussed in Fig 4c and elsewhere, but the data in that very plot seems to suggest that the width of the plateau is shrinking as 1/L (the width of the step for L=120 is about half that for L=60). What is the role of a plateau whose width is going to zero for large system size ?
Is is also not clear to me what controls the boundary between the CO and the IO regions of the phase diagram.
I also don't see the connection between the title "devil's staircase" and the results discussed in the remainder of the manuscript. In my understanding devil's staircase requires a finite stability of each plateau p/q (which might depend on p/q, but not on L->infinity), but this is not the case in the IO region, where the supersolids occur.
Report
I feel the main value of this paper is limited to the numerical observation of the results, but the discussion of the mechanisms at work are difficult to follow and assess. In the current state this manuscript can only be published in SciPost Physics Core.
Carlo Beenakker on 2021-12-26 [id 2055]
[This report is not authored by me but by an independent expert that I consulted as adjudicator.]
I have read the manuscript by Chanda et al, as well as the reports by both referees, including their replies. I believe there is a clash of languages and preconceptions about models and how they should be understood, in particular as it regards to symmetries and the nature of compressible / incompressible phases in this scenario, and how it generalizes to other problems. However, both referees have also pointed out shortcomings in the numerical evidence that have been addressed in this revised version.
Specifically, the criticisms I have seen in those reports regard
(i) All phases in this work are Luttinger liquid, compressible phases (Referee 1) (ii) Conmensurate phases in this work are not evidence of ordered phases (Referee 1) (iii) The wording "incommensurate" / "commensurate" is wrong, because incommensurate phases are compressible (Referee 2) (iv) Lack of evidence of a topological order and fractional excitations in this bosonic system (Referee 2)
The authors have fully addressed points (i) and (ii) by showing that the commensurate order phases do not have long range order and are insulating.
They have addressed point (iii) by illustrating that, even though commensurability is defined in terms of the relation between bosons and spins, and even though there appear staircases, those steps disappear in the incommensurate phase, where the boson density grows continuously in the thermodynamic limit, illustrating the compressibility.
They have also addressed (iv) by providing two further evidences of topological order and fractionalization. One is based on a many-body Berry phase which, similar to earlier works, becomes quantized in the bulk in the topological phase. The other one is based on the referee's own suggestion of that the density of the edge states be measured. The authors show fractionalization of the density in the two almost degenerate edge states, combined with the closing of their gap. I do not find the referee's other arguments regarding degeneracy and topological order convicing because, just as the author note, they are confined to a subset of possible bosonic models.
Overall I find the work is a solid, interesting continuation of earlier investigations by some of the authors. It continues to explore new physics and the emergence many-body phases and topological phenomena in the interplay between a spin and a bosonic model. The interest of the work is fundamental, although the authors also provide a connection to possible implementations with ultracold atoms. This is argubly a bit far from state-of-the-art, but does not detract from the rest of the manuscript, which I believe deserves publication in SciPost.
Further comments:

---

## Round 4 · List of Changes

1. We now show that the off-diagonal correlations $\langle{\hat{b}^{\dagger}_j \hat{b}_{j+R}}\rangle$ decay exponentially with the distance $R$ (Fig. 3 (a) in the updated manuscript) in the commensurate Peierls insulators confirming their insulating nature.
2. The evidence of the existence of diagonal long-range order in the Peierls insulators has been provided by showing that the peak in the structure factor $S_{\sigma}(k)$ remains finite in the thermodynamic limit (Fig. 3 (b) in the updated manuscript).
3. Following Referee’s suggestion, we now refrain from using the term ‘staircase’ when describing the region of Peierls supersolids as this region is compressible and thus the steps in terms of bosonic density $\rho$ vanish in the thermodynamic limit.

You are currently on this page

---

## Editorial Decision

published